# Identification and Validation of Reference Genes for Gene Expression Analysis in *Schima superba*

**DOI:** 10.3390/genes12050732

**Published:** 2021-05-13

**Authors:** Zhongyi Yang, Rui Zhang, Zhichun Zhou

**Affiliations:** 1Research Institute of Subtropical Forestry, Chinese Academy of Forestry, Hangzhou 311400, China; zyyang9266@163.com (Z.Y.); zczhou_risf@163.com (Z.Z.); 2College of Landscape Architecture, Nanjing Forestry University, Nanjing 210037, China; 3Zhejiang Provincial Key Laboratory of Tree Breeding, Hangzhou 311400, China

**Keywords:** reference gene, real-time quantitative PCR, *Schima superba*, tissues, stability evaluation

## Abstract

Real-time quantitative PCR (RT-qPCR) is a reliable and high-throughput technique for gene expression studies, but its accuracy depends on the expression stability of reference genes. *Schima superba* is a fast-growing timber species with strong resistance. However, thus far, reliable reference gene identifications have not been reported in *S. superba*. In this study, 19 candidate reference genes were selected and evaluated for their expression stability in different tissues of *S. superba*. Three software programs (geNorm, NormFinder, and BestKeeper) were used to evaluate the reference gene transcript stabilities, and comprehensive stability ranking was generated by the geometric mean method. Our results show that *SsuACT* was the most stable reference gene and that *SsuACT* + *SsuRIB* was the best reference gene combination for different tissues. Finally, the stable and less stable reference genes were verified using *SsuSND1* expression in different tissues. To our knowledge, this is the first report to verify appropriate reference genes for normalizing gene expression in *S. superba* for different tissues, which will facilitate the future elucidation of gene regulations in this species and useful references for relative species.

## 1. Introduction

Currently, plant gene expression analysis methods include Northern blot, in situ hybridization, RT-PCR, and real-time quantitative PCR (RT-qPCR). RT-qPCR has been widely used in molecular biology research, and expression analysis is realized by real-time detection of fluorescence signal changes in the whole PCR reaction process due to its high sensitivity, accuracy, specificity, throughput capability, and cost-effectiveness [1,2,3,4,5]. However, the accuracy of relative quantification in RT-qPCR is always affected by many variables, such as RNA quality, integrity, reverse transcription efficiency, and amplification efficiency [2,6]. To ensure accurate results and eliminate errors, it is necessary to use one or more stable reference genes to normalize the expression data of target genes [7].

Reference genes, also known as housekeeping genes, refer to a class of genes that can be stably expressed in different tissues and organs. In plant research, the commonly used reference genes are mainly the genes that constitute the cytoskeleton or participate in the cells’ basic biochemical metabolic activities, including *actin* (*ACT*), *β-tubulin* (*TUB*), *ribosomal RNA* (*18S rRNA*, *26S rRNA*), *glyceraldehyde-3-phosphate dehydrogenase* (*GAPDH*), *ubiquitin* (*UBQ*), and *elongation factors 1 α* (*EF-1α*) [8,9,10]. However, studies have shown that the expression levels of these genes are specific and not stable across species, under different treatments. A reference gene suitable for all conditions does not exist [11,12,13]. Therefore, in the qualitative and quantitative research of genes, it is necessary to select the appropriate reference genes based on the specific experimental conditions [14].

*Schima superba* is an evergreen broad-leaved tree in Theaceae, and it is valued commercially for its timber and fire protection [15,16,17]. Theaceae contains about 700 species, which have important economic value, such as the tea plant (*Camellia sinensis*), with its special drinking value, the traditional oil tree *(C. oleifera*), which produces high-quality edible seed oil, and the ornamental plant *C. azalea*, with its attractive flowers. The genus *Schima* has approximately 20 species and is mainly distributed in southern China and the adjacent parts of East Asia, with 13 species (six endemic) present in China [18]. Some reference genes of Theaceae have been reported. For example, *β-actin* could be used as a reference gene for tissues and *GAPDH* for mature leaves and callus in *C. sinensis* [19]. *TUA-3*, *ACT7α*, and *CESA* were relatively stable in the different tissues of six oil-tea *Camellia* spp. [20]. *TUA* and *GAPDH* were optimal reference genes in different organs, as well as *TUB* and *UBQ* in petals in *C. azalea* [21]. However, there have been few reports on gene expression analysis and reference gene expression stability in *S. superba*. Only Yang [22] used *C. oleifera*’s *GAPDH* to verify the differential expression genes between self-and cross-pollinated *S. superba*, but the expression abundance and stability have not been reported. With the completion of genome sequencing and construction of a high-density genetic map [23], molecular design breeding and molecular-assisted breeding of *S. superba* have been carried out gradually. Therefore, it is necessary to select appropriate reference genes for gene expression analysis.

In this study, 19 candidate reference genes, namely, *ColGAPDH*, *SsuACT*, *SsuGAPDH*, *SsuHis* (histone), *SsuTUA1* (α tubulin), *SsuTUA2*, *SsuUBC1* (ubiquitin-conjugating enzyme), *SsuUBC2*, *SsuUBC17*, *SsuUBCJ2*, *SsuMDH* (malate dehydrogenase), *SsuCal7* (calmodulin-7), *SsuCas* (caspase), *SsueIF5* (eukaryotic translation initiation factor 5A4), *SsuMet2* (metallothionin 2a), *SsuGTP* (GTP-binding protein), *SsuRIB* (60S ribosomal protein), *SsuTUB* (tubulin β-3), and *SsuUDP* (UDP-galactose transporter), were assessed by RT-qPCR in different tissues (root, xylem, phloem, leaf, bud, and fruit) of *S. superba*. To obtain the most suitable reference genes, three different statistical tools (geNorm, NormFinder, and BestKeeper) were selected to evaluate the expression stability. In addition, the *SND1* (SECONDARY WALL-ASSOCIATED NAC DOMAIN 1) gene, belonging to the NAC gene family, is involved in the initiation of secondary wall thickening in plant fibroblasts and is the primary switch in the transcriptional regulatory network of secondary wall thickening [24]. To validate the selected best-ranked reference genes, *SsuSND1* expression levels in different tissues were investigated using the most and least stable reference genes or their combination.

## 2. Materials and Methods

### 2.1. Plant Materials

Plant materials were collected from the germplasm bank of *S. superba* clones (119°06′ E, 28°03′ N) in Zhejiang Longquan Academy of Forestry Sciences. The clones were 25-year-old mother trees grafted with scions from Jianou, Fujian (118°31′ E, 27°8′ N) in 2008. The gene bank covers an area of 6.7 hm^2^ and is located at an altitude of 200–300 m in the subtropical monsoon region. The relative humidity of the area is 79%, and the average annual rainfall is 1664.8–1706.2 mm.

In August 2020, different tissues were collected, including secondary xylem, secondary phloem, mature leaf, bud, annual fruit, and root of tissue culture seedlings sub-cultured for 60 days. It was a hot summer, with abundant rainfall and a monthly temperature average of 35 °C. Each sample was set up in biological triplicates. The samples were frozen with liquid nitrogen and stored at −80 °C.

### 2.2. Selection of Candidate Reference Genes

Eighteen candidate reference genes (GeneBank accessions: MW770873–770890) with a stable expression in the different tissues and ages of *S. superba* were selected according to transcriptome data from our laboratory (unpublished, Novogene, Beijing, China) (Appendix A). Eleven of them are often used as housekeeping genes in model plant species. The *ColGAPDH* (*C. oleifera*) (GeneBank accession: KC337052) gene was used as a control [22], and the sequence consistency was only 38.68% with *SsuGAPDH*. According to their CDS sequences blasted from genomic data of *S. superba* (unpublished, Novogene, Beijing, China) (Appendix A), the primers were designed on the web using Primer 3.0 (http://www.primer3plus.com/primer3web/primer3web_input.htm, accessed on August 2020) and synthesized by Sangon Biotech Co., Ltd. (Shanghai, China), as shown in Appendix A.

### 2.3. RT-qPCR Analysis

The total RNA was extracted using the RNAprep Pure Plant Plus Kit (polysaccharides and polyphenolics-rich) (Code No. DP441, TIANGEN, Beijing, China) and was stored at −80 °C. RNA was assessed using 1% agarose gel electrophoresis and quantified with a Nanodrop ND-2000 ultra-micro nucleic acid protein analyzer (Thermo, Waltham, MA, USA). The RNA samples with A260/A280 ratios between 1.8 and 2.1 (Appendix A) were used to synthesize the first strand cDNA with the PrimeScript^TM^ RT master mix (Perfect Real Time) (Code No. RR036A, Takara, Kyoto, Japan) in 20-μL reaction mixtures, once they were adjusted to the same concentration of RNA to 1 μg. The cDNA was diluted 1:9 with nuclease-free water prior to RT-qPCR analysis.

RT-qPCR was restricted to the following guidelines (Applied Biosystems Q7, Waltham, MA, USA): the reaction mixture (20 μL) contained 10 μL of 2× TB GreenPremix Ex TaqII (Tli RNaseH Plus) (Code No. RR820A, TaKaRa, Kyoto, Japan), 2 μL of diluted cDNA, 0.8 μL of each primer (10 mM), 0.4 μL of ROX Reference Dye (50×), and 6 μL of water. The reactions were incubated under the following cycling conditions: 30 s at 95 °C, 40 cycles of 95 °C for 5 s, and Tm 60 °C for 30 s, with a single melt cycle from 65 to 94 °C at 5-s intervals.

The primer specificity was verified by the presence of a single peak in the melt curve analysis during the RT-qPCR process. Three independent biological replicates and three technical repetitions were performed for each of the quantitative PCR experiments. The threshold cycle (Ct) was measured automatically, and correlation coefficients (R^2^) together with slope were calculated from the standard curve based on a tenfold series dilution of the cDNA templates. The corresponding RT-qPCR efficiencies (E) for each gene were determined from the given slope.

### 2.4. Validation of Identified Reference Genes

*SND1* (GeneBank accession: MW796194) was selected as the target gene to validate the reliability of the identified reference genes from transcriptome data (unpublished, Novogene, Beijing, China) (Appendix A). The gene expression profiles at different tissues were normalized using the most and least stable reference gene and *ColGAPDH*. Sample collections and experiments were performed as described above.

### 2.5. Statistical Data Analysis

The average Ct value was calculated from three biological replicates and three technical replicates. Relative gene expression levels were calculated using the 2^−^^△△Ct^ method [25].

GeNorm, NormFinder, and BestKeeper algorithms were used to evaluate the stability of 19 candidate reference genes. GeNorm and NormFinder calculated the average expression stability values based on the 2^−∆Ct^ value [26,27]. BestKeeper calculated the standard deviation (SD), coefficient of variance (CV), and correlation coefficient (r) based on the Ct value [28], using geometric means to provide a comprehensive stability evaluation of candidate reference genes.

## 3. Results

### 3.1. Candidate Reference Genes and PCR Amplification

Eighteen candidate reference genes were selected from the transcriptome of *S. superba*, and *ColGAPDH* was cited from Yang [22]. The presence of a single PCR product of the expected size (Appendix A) and a single peak in the melting curve (Appendix A) confirmed specific amplification. The amplification efficiency (E) of all PCR reactions ranged from 93.47% for *SsuTUA1* to 109.03% for *SsuUBCJ2* (Table 1), suggesting that these genes were suitable for further gene expression analysis. Meanwhile, the standard curves showed good linear relationships, with correlation coefficients (R^2^) above 0.99 (Table 1).

### 3.2. Ct Values of Candidate Reference Genes

To assess the expression stability of 19 candidate reference genes in different tissues, the transcript abundances were presented as their Ct values. The Ct values varied in different tissues, from 16.752 (*SsuMet2*) to 33.379 (*SsuCas*), while the mean Ct values varied from 18.032 (*SsuMet2*) to 25.556 (*SsuMDH*) (Table 2). The Ct range > 4 of *SsuCas*, *SsuUBC17*, *SsuUBC2*, and *SsuUDP* in different tissues suggested that these genes varied greatly and were unstable in different tissues.

### 3.3. Analysis of Reference Gene Stability Using Three Bioinformatic Programs

To reduce analysis error, candidate gene stability ranking in different tissues was determined separately using geNorm, NormFinder, and BestKeeper to screen out the reference genes suitable for experimental treatment and provide a beneficial reference for subsequent research.

The geNorm program was used to rank the gene expression stability by calculating the average expression stability values (M) based on the 2^−∆Ct^ value [26]. The smaller the M value of the reference gene, the more stably it was expressed. Meanwhile, if M > 1.5, it was not suitable as a reference gene [26]. The M values of tested genes evaluated by geNorm are shown in Figure 1. *SsuTUA1* and *SsuRIB* were ranked as the two most stable genes in different tissues, while *SsuCas* and *SsuUDP* were the two least stable genes.

The pairwise variation value (Vn/Vn+1), calculated by geNorm, determined the optimal number of reference genes. When Vn/Vn+1 < 0.15, the optimal number of reference genes is *n*, otherwise, the number is *n*+1 [26]. In this study, except for V18/V19, the other value of Vn/Vn+1 < 0.15 (Figure 2), indicating two reference genes, would be sufficient for gene normalization, and an increase did not improve sensitivity.

NormFinder ranked the expression stability of reference genes by calculating the average pairwise variation in one gene relative to other candidate genes. The smaller the stability value, the more suitable it is as a reference gene [27]. For different tissues, the most stable gene was *SsuACT*, followed by *SsuRIB*, while the least stable gene was *SsuCas*, which was not included with the genes selected by geNorm (Figure 3).

Expression stability is represented by the standard deviation (SD), coefficient of variance (CV), and correlation coefficient (r) of Ct values in the BestKeeper program, and the most stable reference genes were identified as those with the lowest SD and CV and the most r [28]. In this study, *SsuACT* and *SsuUBCJ2* were identified as the most stable genes for different tissues, while *SsuUBC17*, *SsuTUB*, *SsuUBC2*, *SsuUDP*, and *SsuCas* were unstable because of SD > 1 (Table 3).

The rankings of the 19 tested genes were not perfectly consistent among geNorm, NormFinder, and BestKeeper (Table 4). To provide a comprehensive evaluation of candidate reference genes, further analysis was carried out using the geometric mean, which integrates geNorm, Normfinder, and BestKeeper. The comprehensive ranking, recommended by the geometric mean method, is shown in Table 4, and *SsuACT* was the most stable gene for different tissues.

The best combination of reference genes was determined based on the optimal number calculated by geNorm and the ranking list obtained using the geometric mean method. Therefore, *SsuACT* + *SsuRIB* was found to be the best combination of reference genes for different tissues.

### 3.4. Validation of the Identified Reference Genes

To examine the reliability of the candidate reference genes for normalization, *SsuSND1* expression profiles in different tissues were normalized using the two most stable candidate reference genes (*SsuACT* and *SsuRIB*), a combination of stable genes (*SsuACT* + *SsuRIB*), and the least stable reference gene (*SsuCas*), as well as *ColGAPDH* (Figure 4). When *SsuACT*, *SsuRIB*, *SsuACT* + *SsuRIB*, and *ColGAPDH* were used for normalization, the expression patterns of *SsuSND1* were similar, and the relative expression of xylem, leaf, and fruit was higher than the others. *SsuSND1* was hardly expressed in the bud and root, but the expression was most appropriate for *SsuACT* and *SsuACT* + *SsuRIB*. However, as analyzed by *SsuCas*, the expression pattern was not compatible, and the expression levels were too high in the fruit and too low in the bud. It was suggested that the selected reference genes were reliable.

## 4. Discussion

RT-qPCR is a common technique in molecular biology research [29]. In the analysis process, reference genes are often used to reduce or correct the errors in the quantitative process of target genes. Therefore, the selection of an appropriate reference gene is the key to realizing the research of target gene expression under different experimental conditions or tissues [30]. *S. superba* has economic value for its timber, and the wood is used for furniture and construction. According to the phylogenetic analysis of Theaceae, Theeae and Gordonieae are closely related to each other, and the genera Schima belongs to Gordonieae [31]. Moreover, we constructed a high-density genetic map and obtained 168 QTLs for 14 phenotypes [23], but it was not focused on their molecular function or gene expression. Therefore, to carry out the follow-up experiment smoothly, a stable and suitable reference gene would be selected and evaluated for the normalization of gene expression analysis by RT-qPCR in our research.

To avoid the limitations of using only a single software analysis, three bioinformatic programs (geNorm, NormFinder, and BestKeeper) were used to evaluate the expression stability of candidate reference genes in our analysis. The basis for evaluating gene stability in geNorm is the use of each gene’s 2^−∆Ct^ value to calculate the M value [26]. Meanwhile, geNorm can determine the optimal number of reference genes required for quantitative analyses. In this study, gene expression analysis needs two reference genes to achieve the best performance. The NormFinder algorithm is similar to geNorm, using the 2^−∆Ct^ value as the relative expression to calculate the stability of gene expression [27]. BestKeeper focuses on the standard coefficient variation (SD) and variation correlation coefficient (CV) to screen the stability of internal reference genes [28]. The rankings from different programs showed some substantial discrepancies (Table 4). For instance, *SsuTUA1* and *SsuRIB* were the best reference genes identified by geNorm (Figure 1), while *SsuACT* was evaluated as the best by NormFinder (Figure 3) and BestKeeper (Table 3). Differences in rankings among these programs have also been reported in other studies [12,32,33], which are likely the result of the different algorithms that they employ [34]. Therefore, to provide a comprehensive evaluation of candidate reference genes, the geometric mean was used to generate a comprehensive stability ranking, and the best combinations were determined based on the optimal number of reference genes calculated by geNorm. *SsuACT* and *SsuACT* + *SsuRIB* were the most stable reference gene and combination for different tissues (xylem, phloem, leaf, bud, fruit, and root). However, in the study processes, we found that the Ct values of 19 candidate reference genes in seeds were significantly greater than in other tissues, so the transcription levels fluctuated greatly in the seven tissues (root, phloem, xylem, leaf, bud, fruit, and seed). The rankings of the three programs were contradictory; thus, we excluded seeds and analyzed only the remaining six tissues. Notably, *SsuRIB* was the best reference gene predicted among the seven tissues by NormFinder, which indicated that *SsuRIB* could also be used to normalize the gene expression of seeds, because NormFinder is more suitable for the situation when the genes’ transcription level fluctuates greatly [27,28].

Actin is widely used as a reference gene in plants. In this study, *SsuACT* was also the most stable gene. *SsuRIB* and *SsuCal7* are novel genes screened from the genome and transcriptome of *S. superba*, and they have not been reported as reference genes in other species. *ColGAPDH* was not suitable as a reference gene for *S. superba* because of its low expression and poor stability. We compared the reference genes in *S. superba*, *C. sinensis*, *C. oleifera*, and *C. azalea* in Theaceae. The optimal reference genes in different tissues were *SsuACT*, *SsuRIB*, and *SsuTUA1* in *S. superba*; *β-actin* in *C. sinensis* [19]; *TUA-3*, *ACT7α*, and *CESA* in *C. oleifera* [20]; and *TUA* and *GAPDH* in *C. azalea* [21]. Therefore, *ACT* could be used as the reference gene in *S. superba*, *C. sinensis*, and *C. oleifera*, and *TUA* could be used in *S. superba*, *C. oleifera*, and *C. azalea*. Therefore, we presume that *ACT* and *TUA* have wide applicability as reference genes in Theaceae.

When a certain tissue is targeted for investigating gene expression, usually the combination of reference genes should be considered. Hence, to screen the best combination of reference genes for each tissue, we set the moderate Ct value standard to 20–25 and selected the gene with the lowest SD value. The results show that the best combinations of reference genes for each tissue were different, and the best gene combination was *SsuTUA2* and *ColGAPDH* for leaves; *SsuMDH* and *SsuUBC2* for buds; *SsuCal7* and *SsueIF5* for fruits; *SsuCas* and *SsuMDH* for the phloem; *SsuGAPDH* and *SsuGTP* for roots; and *ColGAPDH* and *SsuUBCJ2* for the xylem (Table 2). Differences in the best combination of reference genes between each tissue and different tissues were significant, which indicated that it is necessary to select the appropriate reference genes based on the specific experimental conditions.

To validate the suitability of the identified reference genes, *SsuSND1* expression patterns were investigated in different tissues using different reference genes. The expression patterns normalized by *SsuCas* were not compatible with *SsuACT* + *SsuRIB*. The data once again demonstrate that reference genes play a key role in normalizing the data from RT-qPCR, and the use of inappropriate reference genes may lead to inaccurate results. Moreover, NAC plays a crucial role in the formation and development of the apical meristem [35], lateral root [36], and secondary wall [37]. *SND1* plays a similar role in *S. superba*, and the relative expression of secondary xylem was higher than other tissues.

## 5. Conclusions

As far as our knowledge goes, this study is the first systematic report on the selection and verification of reliable stable reference genes for different tissues in *S. superba*, showing that *SsuACT* was the most stable reference gene, and that *SsuACT* + *SsuRIB* was the best combination for different tissues of *S. superba*. This study provides a basis for gene expression in *S. superba* and Theaceae.

## Figures and Tables

**Figure 1 genes-12-00732-f001:**
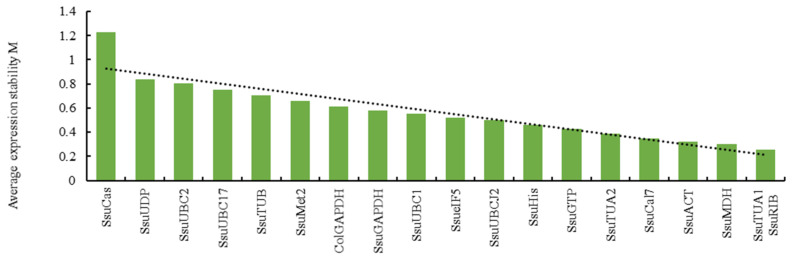
Expression stability values (M) of candidate reference genes calculated by geNorm.

**Figure 2 genes-12-00732-f002:**
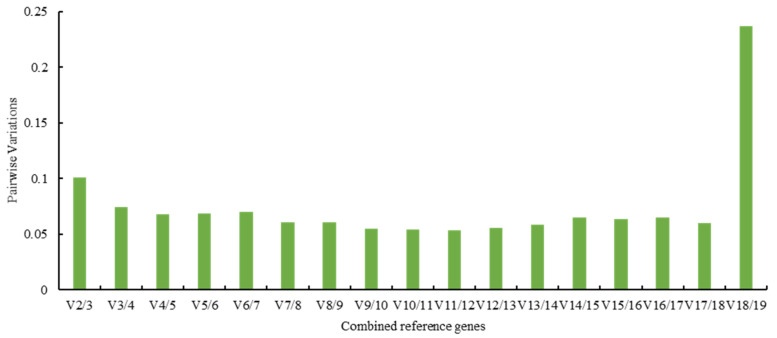
Pairwise variation (V) of candidate reference genes calculated by geNorm.

**Figure 3 genes-12-00732-f003:**
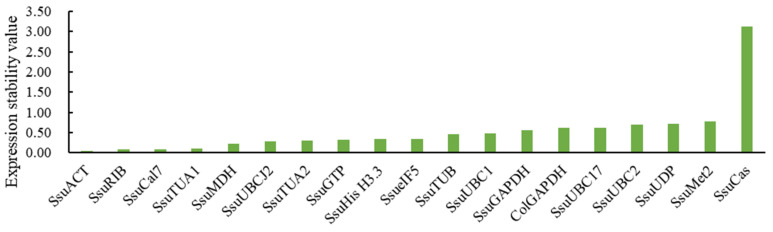
Expression stability of candidate reference genes analyzed by NormFinder.

**Figure 4 genes-12-00732-f004:**
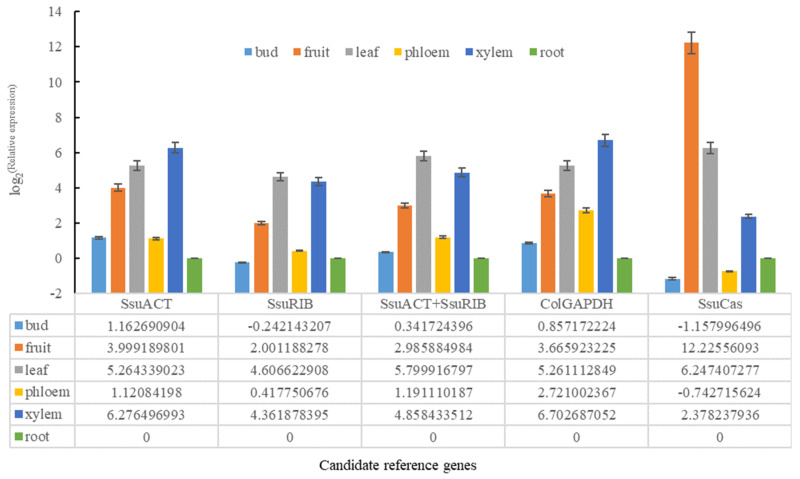
Relative expression of the *SsuSND1* gene using various different reference genes for normalization in different tissues of *S. superba*.

**Table 1 genes-12-00732-t001:** Amplification efficiency and correlation coefficient of candidate reference genes.

Gene	Amplification Efficiency (%)	Correlation Coefficient (R^2^)	Ct Value
*ColGAPDH* [22]	106.14	0.9991	21.562 ± 0.865
*SsuACT*	99.17	0.9981	23.877 ± 1.059
*SsuCal7*	106.74	0.9998	21.984 ± 0.619
*SsuCas*	104.35	0.9977	23.388 ± 1.467
*SsueIF5*	103.76	0.9996	25.185 ± 0.807
*SsuGAPDH*	97.39	0.9996	20.608 ± 0.903
*SsuGTP*	100.02	0.9929	23.709 ± 4.902
*SsuHis*	109.7	0.9991	21.455 ± 1.150
*SsuMDH*	102.6	0.9994	20.113 ± 0.546
*SsuMet2*	97.27	0.9995	22.875 ± 0.808
*SsuRIB*	94.92	0.9931	22.073 ± 1.003
*SsuTUA1*	93.47	0.9991	25.556 ± 1.004
*SsuTUA2*	104.27	0.9986	18.032 ± 0.984
*SsuTUB*	103.17	0.9984	22.314 ± 0.848
*SsuUBC1*	105.46	0.9907	25.320 ± 0.817
*SsuUBC17*	107.89	0.9993	21.644 ± 1.555
*SsuUBC2*	104.03	0.9985	21.556 ± 1.681
*SsuUBCJ2*	109.03	0.9975	24.512 ± 1.180
*SsuUDP*	108.71	0.9989	24.877 ± 1.782

Ct (cycle threshold) means the number of cycles experienced when the fluorescent signal in each reaction tube reaches the set domain value; SD: standard deviation. Ct values are mean ± SD (*n* = 3).

**Table 2 genes-12-00732-t002:** Average Ct (cycle threshold) values ± SD (standard deviation) of candidate reference genes in different tissues of *Schima superba*.

Gene	Leaf	Bud	Fruit	Phloem	Root	Xylem	Average	Min	Max	Range
*SsuACT*	21.125 ± 2.752	20.818 ± 0.668	22.681 ± 1.065	21.606 ± 0.604	22.504 ± 1.687	20.638 ± 0.700	21.562	20.638	22.681	2.043
*SsuTUA1*	23.116 ± 3.300	22.855 ± 1.102	24.844 ± 1.528	23.801 ± 0.875	25.474 ± 0.979	23.173 ± 0.766	23.877	22.855	25.474	2.62
*SsuTUA2*	21.717 ± 1.408	21.637 ± 1.138	22.489 ± 1.343	21.927 ± 0.284	22.918 ± 1.060	21.214 ± 0.388	21.984	21.214	22.918	1.704
*SsubTUB*	23.802 ± 2.168	22.222 ± 0.634	24.350 ± 0.855	22.667 ± 0.281	25.599 ± 1.185	21.687 ± 0.812	23.388	21.687	25.599	3.911
*ColGAPDH*	24.569 ± 1.992	24.188 ± 1.022	26.065 ± 1.048	26.214 ± 0.373	25.079 ± 0.822	24.997 ± 0.272	25.185	24.188	26.214	2.026
*SsuCal7*	20.469 ± 2.372	19.505 ± 0.960	21.703 ± 0.636	20.858 ± 0.228	21.444 ± 1.328	19.672 ± 1.033	20.608	19.505	21.703	2.198
*SsuCas*	22.702 ± 3.140	20.807 ± 0.850	33.379 ± 1.396	21.377 ± 0.187	23.713 ± 2.040	20.280 ± 0.637	23.709	20.28	33.379	13.099
*SsueIF5*	21.820 ± 2.500	19.810 ± 1.122	21.700 ± 0.709	21.423 ± 0.266	23.243 ± 1.744	20.736 ± 0.619	21.455	19.81	23.243	3.433
*SsuGAPDH*	19.325 ± 2.828	19.818 ± 1.308	20.596 ± 0.941	20.723 ± 0.325	20.409 ± 0.326	19.810 ± 0.278	20.113	19.325	20.723	1.398
*SsuGTP*	22.843 ± 2.547	21.975 ± 1.044	22.920 ± 1.006	22.966 ± 0.437	24.304 ± 1.194	22.239 ± 0.692	22.875	21.975	24.304	2.328
*SsuHis*	21.538 ± 2.536	20.623 ± 0.710	22.414 ± 0.705	22.775 ± 0.268	23.434 ± 2.293	21.654 ± 0.458	22.073	20.623	23.434	2.811
*SsuMDH*	24.656 ± 3.161	24.456 ± 0.432	26.864 ± 1.025	25.655 ± 0.211	26.601 ± 0.786	25.104 ± 0.760	25.556	24.456	26.864	2.408
*SsuMet2*	17.014 ± 2.966	16.752 ± 0.789	17.975 ± 1.264	19.275 ± 0.189	18.652 ± 1.393	18.522 ± 0.908	18.032	16.752	19.275	2.523
*SsuRIB*	21.848 ± 2.851	21.358 ± 0.871	23.163 ± 1.038	22.122 ± 0.454	23.539 ± 2.217	21.856 ± 0.544	22.314	21.358	23.539	2.181
*SsuUBC1*	24.886 ± 3.431	24.180 ± 0.886	25.768 ± 0.564	26.410 ± 0.186	25.820 ± 0.990	24.857 ± 0.396	25.32	24.18	26.41	2.229
*SsuUBC17*	22.491 ± 3.106	20.030 ± 1.158	21.798 ± 1.360	21.316 ± 0.288	24.137 ± 2.867	20.090 ± 0.560	21.644	20.03	24.137	4.107
*SsuUBC2*	22.343 ± 2.593	20.277 ± 0.247	21.973 ± 1.268	20.856 ± 0.569	24.277 ± 2.711	19.607 ± 0.508	21.556	19.607	24.277	4.67
*SsuUBCJ2*	24.718 ± 2.937	23.014 ± 0.667	24.883 ± 0.758	24.601 ± 0.248	26.375 ± 2.294	23.482 ± 0.350	24.512	23.014	26.375	3.361
*SsuUDP*	25.303 ± 3.170	23.309 ± 1.093	25.399 ± 0.965	24.038 ± 0.669	27.973 ± 2.695	23.242 ± 0.684	24.877	23.242	27.973	4.73

**Table 3 genes-12-00732-t003:** Expression stability of candidate reference genes analyzed by BestKeeper.

Gene	Geometric Mean	Average Mean	Minimum	Maximum	SD	CV	r	*p*-Value
*SsuACT*	21.55	21.56	20.64	22.68	0.7	3.25	0.987	0.001
*SsuUBCJ2*	24.49	24.51	23.01	26.38	0.84	3.44	0.987	0.001
*SsuCal7*	20.59	20.61	19.51	21.7	0.73	3.53	0.977	0.001
*SsuHis*	22.05	22.07	20.62	23.43	0.8	3.63	0.93	0.007
*SsuRIB*	22.3	22.31	21.36	23.54	0.69	3.1	0.928	0.008
*SsuUDP*	24.83	24.88	23.24	27.97	1.35	5.42	0.926	0.008
*SsuTUA1*	23.86	23.88	22.85	25.47	0.85	3.58	0.92	0.009
*SsuUBC17*	21.6	21.64	20.03	24.14	1.17	5.38	0.919	0.01
*SsuTUA2*	21.98	21.98	21.21	22.92	0.48	2.18	0.905	0.013
*SsuUBC2*	21.5	21.56	19.61	24.28	1.31	6.07	0.887	0.018
*SsuTUB*	23.35	23.39	21.69	25.6	1.2	5.11	0.858	0.029
*SsuCas*	23.35	23.71	20.28	33.38	3.22	13.6	0.83	0.041
*SsuMDH*	25.54	25.56	24.46	26.86	0.82	3.2	0.824	0.044
*SsueIF5*	21.43	21.46	19.81	23.24	0.8	3.72	0.761	0.079
*SsuGTP*	22.86	22.87	21.98	24.3	0.52	2.28	0.742	0.092
*SsuUBC1*	25.31	25.32	24.18	26.41	0.68	2.68	0.741	0.092
*SsuGAPDH*	20.11	20.11	19.32	20.72	0.46	2.3	0.634	0.176
*ColGAPDH*	25.17	25.19	24.19	26.21	0.64	2.53	0.602	0.206
*SsuMet2*	18.01	18.03	16.75	19.27	0.78	4.35	0.513	0.296

SD: standard deviation; CV: coefficient of variance; r: correlation coefficient.

**Table 4 genes-12-00732-t004:** Comprehensive evaluation of stability of candidate reference genes.

Gene	geNorm	NormFinder	BestKeeper	Geometric Mean	Combined Ranking
*SsuACT*	3	1	1	1.44	1
*SsuRIB*	1	2	5	2.15	2
*SsuTUA1*	1	4	6	2.88	3
*SsuCal7*	4	3	3	3.3	4
*SsuMDH*	2	5	8	4.31	5
*SsuUBCJ2*	8	6	2	4.58	6
*SsuTUA2*	5	7	7	6.26	7
*SsuHis*	7	9	4	6.32	8
*SsuGTP*	6	8	10	7.83	9
*SsueIF5*	9	10	9	9.32	10
*SsuUBC1*	10	12	11	10.97	11
*SsuGAPDH*	11	13	12	11.97	12
*ColGAPDH*	12	14	13	12.97	13
*SsuTUB*	14	11	16	13.51	14
*SsuMet2*	13	18	14	14.85	15
*SsuUBC17*	15	15	15	15	16
*SsuUBC2*	16	16	17	16.33	17
*SsuUDP*	17	17	18	17.33	18
*SsuCas*	18	19	19	18.66	19

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
