# Peer review of "Identification and Validation of Reference Genes for Gene Expression Analysis in *Schima superba"

_genes, 2021, doi:10.3390/genes12050732_

Round 1
Reviewer 1 Report
Line 95 S. superba must be italic
Line 100 S. superba must be italic
Line 142 S. superba must be italic
Line 276 S. superba must be italic
Line9 - please remove 'Background:' word Line13 - please remove 'Result:' word Line19 - please remove 'Conclusions:' word Line26 - Please use 'Currently' besides 'At present'. Line32 - Please use 'Such as' besides 'for example'. The authors explain in more detail why these genes are used as a reference in the introduction part. The authors should add a part related to bioinformatic analysis in the material method part and they have to explain and give detail about bioinformatic analysis. The authors should compare 19 reference genes with other species results in discussion part.Author Response
- In lines 102, 108, 154, and 298, superba has been changed to italic.
- In lines 10, 14, and 20, “Background,” “Result,” and “Conclusions,” respectively, were removed.
- In line 29, “Currently” was used instead of “At present.” In line 35, use “such as” was used instead of “for example.”
- We have added a paragraph related to bioinformatic analysis in “Materials and Methods” (lines 146–151). “GeNorm, NormFinder, and BestKeeper algorithms were used to evaluate the stability of 19 candidate reference genes. GeNorm and NormFinder calculated the average expression stability values based on the 2−∆Ct value [26–27]. BestKeeper calculated the standard deviation (SD), coefficient of variance (CV), and correlation coefficient (r) based on the Ct value [28], using geometric means to provide a comprehensive stability evaluation of candidate reference genes.”
- In the “Discussion” section (lines 279–290), we compared the selected reference genes with those of Theaceae plants ( sinensis, C. oleifera, and C. azalea). Because they belong to Theaceae, are distributed in subtropical areas, and possess important economic value, they have similarities in genetic evolution and living environment; therefore, it is more meaningful to compare their reference genes.
Reviewer 2 Report
This manuscript describes the identification and validation of appropriate reference genes
for gene expression analysis in Schima superba. They concluded that SsuACT is the most stable reference gene and SsuACT + SsuRIB was the best combination for different tissues.
There seem to be several points to be improved and clarified.
- ln 91: As RT-qPCR depends on the sampling season or other growing conditions, they should be described in detail.
- ln 198: Table 4 presents a comprehensive evaluation of the stability of candidate reference genes using geometric mean and suggests their combined ranking. However, in a practical view, if a certain tissue is targeted for investigating gene expression, usually the combination of candidate reference genes should be considered.
I think the author can show additionally the best combination of candidate reference genes for each tissue, although SsuACT + SsuRIB was the best combination for different tissues.
- This manuscript determines candidate reference genes in the different tissues, but in fact, the reference genes depend on biotic and abiotic stress conditions. According to my point of view, it should be done at different stress conditions.
[Other minor corrections]
- ln 42: grammer correction is needed
- ln 180: -> a reference gene
- ln 185: -> represented
- ln 200: -> the geometic
- ln 206: -> the combination
- ln 213: -> the bud
- ln 231: -> a single
- ln 234: consider rephrasing: use each gene's 2-delta Ct value to calculate the M value
- ln 245: the sentence could be improved
- ln 247: a comprehensive
- ln 250: combinations
- ln 258: -> when the genes'
- ln 259: -> Actin is widely used as a reference gene
- ln 260: -> the genome
- ln 271: a dangling expression: consider rephrasing
- ln 275: key -> crucial
Author Response
- In lines 83–93, in the Plant materials sub-section, we introduced the plant growth environment (subtropical monsoon area, altitude, relative air humidity, annual rainfall, and so on), as well as the temperature and precipitation of the sampling season.
- You indicated that we should show the best combination of candidate reference genes for each tissue.
- We screened out the reference gene combinations suitable for different tissues, which had a wide range of applicability.
- At present, the software programs for screening reference genes requires no fewer than two samples; therefore, it was impossible to screen only a tissue’s reference genes through this common method.
- As a reference gene, it required moderate and stable expression. Therefore, I hoped to select the reference gene with a minimal SD value and moderate Ct value, but I could not determine the standard of what was considered moderate. Hence, we set the standard to 20–25 and selected the gene with the lowest SD value. The results are as follows: the best gene combination were SsuTUA2 and ColGAPDH for leaves; SsuMDH and SsuUBC2 for buds; SsuCal7 and SsueIF5 for fruits; SsuCas and SsuMDH for the phloem; SsuGAPDH and SsuGTP for roots; and SsuColGAPDH and SsuUBCJ2 for the xylem. The results have no regularity and contradict the results of the three software packages; thus, we do not think these results are convincing.
- The reference genes depend on biotic and abiotic stress conditions. We only determined reference genes in the different tissues because we focus on the tissue development of superba because it is a high-quality timber species. In the future, we may do some stress research as a new direction.
- I have corrected the grammatical errors.
Round 2
Reviewer 2 Report
The following conclusive sentence should be included in the conclusion or abstract; "The best gene combinations were SsuTUA2 and ColGAPDH for leaves; SsuMDH and SsuUBC2 for buds; SsuCal7 and SsueIF5 for fruits; SsuCas and SsuMDH for the phloem; SsuGAPDH and SsuGTP for roots; and SsuColGAPDH and SsuUBCJ2 for the xylem."
The problems in English grammar were very improved by massive editing.
Author Response
We have added the conclusive sentence in the discussion section (Lines 275-285). “When a certain tissue is targeted for investigating gene expression, usually the combination of reference genes should be considered. So to screen the best combination of reference genes for each tissue, we set the moderate Ct value standard to 20–25 and selected the gene with the lowest SD value. The results showed that the best combinations of reference genes for each tissue were different, and the best gene combination was SsuTUA2 and ColGAPDH for leaves; SsuMDH and SsuUBC2 for buds; SsuCal7 and SsueIF5 for fruits; SsuCas and SsuMDH for the phloem; SsuGAPDH and SsuGTP for roots; and ColGAPDH and SsuUBCJ2 for the xylem (Table 2). Differences in the best combination of reference genes between each tissue and different tissues were significant, which indicated that it is necessary to select the appropriate reference genes based on the specific experimental conditions.” And added the SD values in the Table 2.
We did not add the conclusive sentence in the conclusion or abstract. Because we think the “moderate Ct value standard to 20–25” has no resources to refer to, and it should be not stationary. Therefore, for the sake of conservatism, we choose to put it in the discussion section.